# Patients’ Preferences and Expectations in Overactive Bladder: A Systematic Review

**DOI:** 10.3390/jcm12020396

**Published:** 2023-01-04

**Authors:** Antonio Cicione, Riccardo Lombardo, Vincenzo Umbaca, Giorgia Tema, Giacomo Gallo, Jordi Stira, Carmen Gravina, Beatrice Turchi, Antonio Franco, Elisa Mancini, Antonio Nacchia, Rocco Damiano, Andrea Tubaro, Cosimo De Nunzio

**Affiliations:** 1Urology Unit, Ospedale Sant’Andrea, ‘Sapienza’ University of Rome, 00191 Rome, Italy; 2Urologia, ASP Catanzaro, 88100 Catanzaro, Italy; 3Unit of Urology, Magna Graecia University of Catanzaro, 88100 Catanzaro, Italy

**Keywords:** overactive bladder, OAB, preferences, expectations

## Abstract

The aim of our study is to review the current available knowledge regarding preferences and expectations of patients with overactive bladder (OAB). The Preferred Reporting Items for Systematic Reviews and Meta-analyses (PRISMA) statement guidelines were followed for this manuscript’s preparation. Three online databases were searched: PubMed/Medline, Embase, and Scopus, while a combination of the following keywords was used: detrusor overactivity, overactive bladder, urinary incontinence, perspectives, expectations, and preferences. Overall, 1349 studies were retrieved and screened while only 10 studies appeared to be relevant for the scope of this review. Most of the studies were related to preferences about OAB medications (i.e., antimuscarinics); four of them reported patients’ inclinations to alternative treatments in the case of medication therapy failure (i.e., neuromodulation, Botox). No data were found about diagnosis or other aspects of disease management (i.e., surgery, follow-up). Based on these findings, from the patient’s point of view, the ideal medication should be cheap, without risk of cognitive function impairment, and able to reduce daytime urinary frequency and incontinence episodes.

## 1. Introduction

Overactive bladder (OAB) is defined as urgency, with or without urgency urinary incontinence, usually with increased daytime frequency and nocturia, with no proven infection or other obvious pathology [1]. OAB is a descriptive clinical term, a symptom-based definition, and it is not diagnostic of a specific disease. The pathophysiology of OAB is poorly understood, but probably involves changes at multiple levels of micturition control (i.e., brain, spinal cord, and smooth muscle of the bladder). However, the true aetiology of OAB is still unknown.

Treatment of OAB is very challenging for several reasons. First, about one patient over two with bothersome OAB symptoms consulted a physician and less than a quarter received treatment [2]. Thus, several treatments have been proposed for OAB including lifestyle changes (first line treatment), medical and surgical options. Regarding medical options, several antimuscarinics agents are widely used for OAB symptoms. However, their fair efficacy and side effects often result in a poor adherence. More specifically, adherence rates at one year may be as low as 35% [3]. Surgical options are usually reserved for refractory cases or in presence of OAB related complications. Nonetheless, surgical options need to be accurately tailored to the patient’s bladder condition, overall medical status, physical abilities, and expectations.

Nowadays, patient-centred care and shared decision making (SDM) are generally recognized as the gold standard for medical consultations. SDM is an approach that involves a mutual discussion regarding management or treatment options to identify the best option for the patient in terms of risk-benefit ratio and patient’s preferences. Likewise, from the clinician’s perspective, SDM is a useful way of presenting to patients their health condition. Some authors have suggested that treatment adherence could be improved by enhancing the interaction between physician and patient and cost ‘effectiveness [4,5].

Furthermore, there is an increasing interest to include patients’ preferences into recommendations for diseases management. For instance, the National Institute for Clinical Excellence (NICE) has developed a research project to test various methods of surveying patient preferences. The current guidelines from the European Association of Urology include the participation of patient representatives among the Authors.

With this knowledge in mind, aim of our study was to systematically review the current knowledge on expectations and preferences of patients with OAB.

## 2. Evidence Acquisition

The Preferred Reporting Items for Systematic Reviews and Meta-analyses (PRISMA) statement guidelines were followed during manuscript preparation of this review. A protocol was developed and approved in the PROSPERO database (CRD: 42022327200).

Literature research was carried out in September 2022 to identify published studies relevant to the scope of this review. Three online databases were searched: in PubMed/Medline, Embase and Scopus while a combination of the following keywords was used: detrusor overactivity, overactive bladder, urinary incontinence, perspectives, expectations, and preferences. The reference lists of all manuscripts reviewed as full text were also screened for eligible studies. Two independent authors (AC, RL) screened the databases and disagreements were resolved upon consensus with a senior author (CDN). PRISMA flow chart is available in Figure 1.

Studies were included if published in English and available online while they were excluded if related to children population (age < 18 years), faecal incontinence, post radical prostatectomy incontinence, and if patients’ preferences were not the outcome. This research strategy yielded 1349 papers with potentially relevant title and abstract. However, only ten studies were deemed to be relevant for the purpose of this review (Figure 1) after screening. No previous reviews were found.

### Risk of Bias Assessment

The risk of bias of each included study was assessed by two review authors, working independently using the Newcastle-Ottawa scale for observational studies [6].

## 3. Evidence Synthesis

### 3.1. How to Evaluate Patients’ Preferences and Expectation?

Several tools are described as possible methods to evaluate and compare patients’ preferences and expectations. Quantitative data were collected using dedicated questionnaires. Overall, seven studies used a discrete choice experiment, two studies used oral interviews, and one study used questionnaires with a utility score.

Regarding DCE, this is a quantitative technique for eliciting individual preferences. It allows researchers to discover how individuals value selected attributes of a programme, product, or service by asking them to state their choice over different hypothetical alternatives [7]. DCE is also known as a conjoint analysis and usually the choice set is composed of two or more competing alternatives which vary in terms of several attributes. In recent years, DCEs have been increasingly used to help understand preferences in the field of health and healthcare. Interestingly, one study [8] used health utility index as an approach to compute the value that patient assigned to some aspects of OAB, including both therapy and symptoms severity [8]. The Utility index was originally developed as a rating scale to measure general health status and health-related quality of life (HRQoL). However, it seems a simple way to assign and compare values between different contexts.

The retrieved studies highlight the absence of validated and standardized tools to assess and measure patient’s preferences and expectations regarding a treatment option or a therapeutic strategy. This limit probably depends on differences in cultural, geographical, society, economic, and national factors [9].

Although there is no world-wide validated tools to assess patients’ preferences and expectations. Previous experiences on the use of DCE in other areas such as the evaluation of environmental goods and services, indicate that we can be optimistic on the utility of this approach. Moreover, Sumedha Chhatre et al. (2021) recently showed that the use of DCE [10] for assessing OAB preferences is feasible and acceptable whereas future research remains required to evaluate its use in different subgroups distinct for sociodemographic and clinical characteristics [11,12,13].

### 3.2. Patients’ Preferences and Expectations in Medical Treatment

Overall, eight studies addressed the preferences and expectations of OAB patients: four studies were related to the use antimuscarinics, four studies included patients undergoing treatments for OAB unresponsive to antimuscarinics, and one study evaluated the importance of economic burden in patient’s preferences (Table 1).

Amod Athavale et al. (2018) evaluated medical treatment preferences of treatment-naïve individuals with symptoms of OAB [14]. Data from 514 patients were analysed, 68% of them were female, and 66% of the cohort were less than 65 years of age. Most respondents reported moderate/severe OAB (64.2%) and experienced incontinence (79.4%) and/or nocturia (59.1%). A DCE was designed to quantify the strength of preference. This survey showed that treatment-naïve patients assigned a strong preference for drug delivery method, reduction of daytime micturition frequency, and lower costs. More specifically, oral and patches were more desirable than injectable therapies. Furthermore, the study highlighted how physician and patient perspectives may be different. Physicians are mostly focused on efficacy, whereas patients on tolerability [15,16,23].

Overall, patients on AMs present a higher risk of side effects when compared to placebo [17,24] (1.26 times). The study evaluated frailly elderly patients’ perspectives about AEs of AMs treatment. DCE analysis highlighted how much older patients are concerned by cognitive side effects of AMs treatment.

OAB treatment can be expensive for the patient. Harpe et al., in a survey based on 133 OAB patients, investigated medication preferences using a dedicated questionnaire including nine hypothetical scenarios [18]. Briefly, participants were asked to provide an estimate of the likelihood (from 0% to 100%) of medications to control OAB symptoms vs. doing nothing to control symptoms. Insurance coverage was the most important attribute for choosing a treatment, followed by sleep disturbance [18].

### 3.3. Patients Preferences and Expectations in Minimally Invasive Treatment

Patients nonresponsive to medical treatment may benefit from 3rd line treatment including sacral neuromodulation (SNM), onabotulinum toxin A (Botox^®^), and/or percutaneous tibial nerve stimulation (PTNS). Hashim Hashim (2015) investigated patient preferences for these treatments and their respective characteristics in a sample of patients with idiopathic OAB in the UK. Among 127 (91%) of patients included in the analysis, most of them (≥80%) were willing to try each of the three treatments: respectively, 57%, 34%, and 9% most preferred PTNS, SNM, and Botox. Furthermore, preferences for the attributes differed from each treatment the patients chose. For instance, patients choosing SNM favoured an implanted device in upper buttock more than those preferring PTNS or Botox. Moreover, it has been recently reported that successful rate of SNM improves when the patient can interact with the device [22] while low side effects followed by the effectiveness on continence and micturition frequency had the greatest impact on patients when a SNM device was implanted [20]. Moreover, Fontaine et al. evaluated 217 patients who failed medical treatment. Nobody opted for ileal conduit. Further, 25% opted for Botox injections, and 25% for SNM. Only 2.5% of patients chose cystoplasty ± Mitrofanoff channel reconstruction. 

## 4. Conclusions

Our review summarizes the current evidence on patients’ preferences and expectations regarding OAB management (Table 2). Unfortunately, few studies investigated this topic using different non-standardized methods.

Patient preferences and expectations should be part of the decision making process in managing benign and malignant disorders [25,26,27,28,29,30]. Recently, Malde et al. summarized patient preferences and expectations in LUTS management and clearly showed how patient and physician perspectives may be completely different and that often patients prefer a less effective treatment but with a minor risk of AEs [31,32,33]. Our findings in a similar topic confirm these results. In OAB patients, medical treatment is a challenge scenario for the physicians considering the low adherence and satisfaction to medications. Several explanations have been proposed, such as lack of efficacy, rate of adverse events, costs, and patients’ awareness on their condition. A better knowledge of patients expectation and preferences could help physicians to better profile OAB patients in order to identify the best treatment for every single patient, to improve patient adherence and compliance and to reduce doctor ‘google’ and doctor shopping [25,34].

With respect to the current evidence, OAB patients prefer an oral treatment which the reduces urgency, frequency, and incontinence episodes, with no effect on cognitive function and covered by insurance. The impact of costs and insurance coverage on patients’ compliance is particularly evident in Europe, where in some countries AMs and beta-3-agonists are not covered and as a consequence patients adherence and compliance is lower when compared to other countries.

The lack of standardized method in the evaluation of patient’s preferences and expectations represents an important bias of this research area. Given the poor evidence on the subject, first of all, it is very important to develop a validated and universal method to assess patients’ preferences and expectations. This approach could lead to a better understanding of patient’s preferences and expectations, better analysis, and identifying possible differences based on social and demographic characteristics. Another unmet need in OAB is that most of the available evidence is based on female population.

Currently, the importance of sharing decisions on treatment with patients has long been recognized and this will lead to consider a “patient-based approach” instead of “one size fits all” strategy.

## Figures and Tables

**Figure 1 jcm-12-00396-f001:**
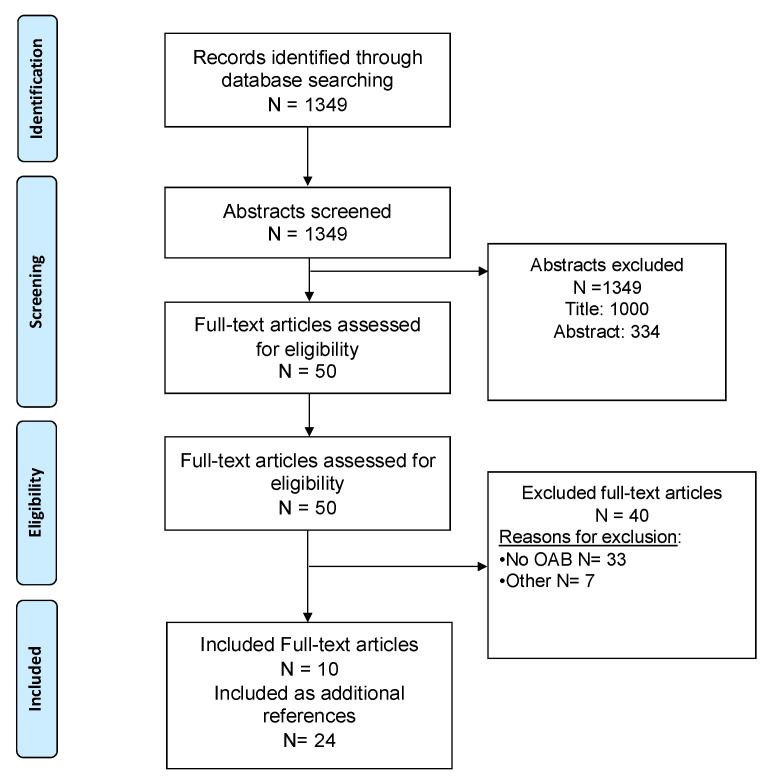
Preferred Reporting Items for Systematic Reviews and Meta-Analyses (PRISMA) Flow diagram of studies identified, excluded and included.

**Table 1 jcm-12-00396-t001:** Studies included in the systematic review.

Authors	Year	Assesment Method	Evaluated Outcome	Main Findings
**Amod Athavale** [14]	2018	Internet DCE	Pharmacotherapy treatment preferences in treatment-naïve US individuals with symptoms of OAB	Strong preference for oral and patches over injectable therapies, reduction daytime micturition frequency and lower out-of-pocket costs
**M. Heisen** [15]	2016	DCE	Patient and physician preferences for oral pharmacotherapy for OAB in five European countries	physicians put more emphasis on higher benefits, while patients on limiting risks of side effects. Both groups valued incontinence as the most important attribute
**P. Swinburn** [16]	2010	DCE	To grade preferences for benefit and side effects of antimuscarinics	Incontinence, followed by urgency, micturition, constipation and dry mouth were found to be the attributes significantly influencing treatment preference
**Veerle H. Decalf** [17]	2017	DCE	To examine the importance attributed by older people (median age 75 yrs) to the most side effects of antimuscarinics	The most unwanted side effect in the choice of antimuscarinics for OAB was severe cognitive effect
**Spencer E. Harpe** [18]	2007	DCE	Assessment of importance of economic burden for OAB therapy	Presence of prescription drug insurance is the most important factor to patients considering the treatment of OAB symptoms.
**Hashim Hashim** [19]	2015	DCE	Patient preferences for refractory OAB treatmentsPercutaneous tibial nerve stimulation (PTNS)Botulinum toxin (Botox^®^) Sacral neuromodulation (SNM)	127 pts were respectively willing to try PTNS (56.7%), SNM (34%) and Botox (9.4%) as next treatment option
**PMH Sanders** [20]	2011	DCE	Preferences on different neural prostheses	Side effects had the greatest significant impact on subject choices, followed by the effectiveness on continence and voiding
**Jennifer M. Wu** [8]	2011	Questionnaires with a utility score	Compute utility index for several OAB aspects including treatment and symptoms severity	Moderate or severe symptoms, as being quite burdensome. The degree of invasiveness and the number of adverse effect/complications are important contributors to assign utility index to the various treatment options
**Christina L. Fontaine** [21]	2017	interview	Which treatment patient choose in case of medical therapy failure	On 217 patients, nobody opted for ileal conduit. 25% opted for Botox injections and 25% for SNM. Only 2.5% of patients chose cystoplasty ± Mitrofanoff channel reconstruction.
**Manon te Dorsthorst** [22]	2021	Cohort study	Factors predicting success of neuromodulation treatment	Patient’ interaction with device ameliorates successful rate

**Table 2 jcm-12-00396-t002:** Patients’ preferences and expectations regarding OAB management.

	Medical Treatment	Invasive Treatment
Patient preferences	An oral treatment which reduces urgency, frequency and incontinence episodes, with no effect on cognitive function and covered by insurance.	- 3rd line treatments (i.e., sacral neuromodulation (SNM), onabotulinum toxin A (Botox^®^), and/or percutaneous tibial nerve stimulation (PTNS)) are preferred over ileal conduit. - Patient decision is influenced by device interactivity, effectiveness on continence and micturition frequency.

## Data Availability

Data will be available upon request.

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
