# Peer review of "Patients’ Preferences and Expectations in Overactive Bladder: A Systematic Review"

_jcm, 2023, doi:10.3390/jcm12020396_

Round 1
Reviewer 1 Report
Interesting topic and well-researched. Would benefit from displaying results in a graphical form, or using some other visual aid to help readers digest results.
Author Response
REVIEWER 1
Interesting topic and well-researched. Would benefit from displaying results in a graphical form, or using some other visual aid to help readers digest results.
Reply. We thank the reviewer for his/her suggestion. A new table (Table 2) was made with the aim to summarize the findings about patients’ preference regarding OAB treatments.
Table 2. Summary of patients’ preferences and expectations regarding OAB management
|
Medical treatment |
Invasive treatment |
Patient preferences |
An oral treatment which reduces urgency, frequency and incontinence episodes, with no effect on cognitive function and covered by insurance. |
- 3rd line treatments (i.e.: sacral neuromodulation (SNM), onabotulinum toxin A (Botox®), and/or percutaneous tibial nerve stimulation (PTNS)) are preferred over ileal conduit. - Patient decision is influenced by device interactivity, effectiveness on continence and micturition frequency. |
Reviewer 2 Report
The power of the paper is the usage of multiple studies with big data in different ways. Thus several tools are used with different modes and techniques also looking at different health cares Data surveying patient preferences from NICE.
The preferring Reorting Items for sytemtic and Metananlyses (Prisma) statement guidelines were used followding the manuscript of this review. A protocol was developed and aproved in the PROSPER Database (CRD: 42022327200) Literture search was carried out in September 2022 to idenifiy piblished studies MedMedline, Ambase and Scopus relevant detrurosor acvtivity, ovaractive bladder, urinary incontinence , perspectives, expectationsand preferences. 2 independent authors screened the databses and diesgreementswere resolved upon consensus with a senior author scopeing the review.
Studies were included if published in English and available online but excluded if related to children population fecal incontinence postradical post prostatectomy in patients incontinence and in case were not the outcome.
The research yielded 1349 papers.
Evidence Synthesis asked for evaluation of patients´ prefreneces and ecpetation.Qulittative data evaluated and compared patients preferences and excpeectationsOverall seven studies used a discrete choice system.
Patients preferences and expectations in medical treatment
8 studies adressed the the prefrences and expectations of OAB patients, 4 related the use of antimuscarinics, 4 related the treatment unresponsive to antimuscarinics . One study evaluated the importance of economic burden in patinet´s preferences.The retrieved studies highlight the absence of validated and standradized toolsmeasure pts preferences .
1 study evaluated the treatment preferences individuals with symptoms.Treatment-naive pts.
Looking at the data currently published shows most of the studies using different non-standardized methods. This may be an issue to change.
Patient preferences and excpecations should be a part of the decision making process in benign and malignant disoreders. Malde summarizedpatients preferences and expectations in LUTS management and clearly showed the differenec betwenn doctors and patient wishes and showing that patient were in favour of less effective treatment with minor risk. This study shows a similar confirms this observation.
Several explanations hav ebeen shown as a lack of efficacy, rare adeverse events and dpatient awareness of on their condition. A better knowledgeof patinet expectation and preferences can help better help physician to identify the best single patient, to improve patient adherence and compliance.
OAB patinet prefere oral medicationreucing urgency, frequency, and incontinence episodeswithout effect of cognitove function and coveres insurance limiting patienta dherence and compliance.
Thelack of standardized methods on evaluation reprensent and an important bias in that research area.
Sharing desissions on treatment is a corner stone of Treatment sucess, adherence and willingness to treamtne in OAB patientes.
Author Response
REVIEWER 2
The power of the paper is the usage of multiple studies with big data in different ways. Thus several tools are used with different modes and techniques also looking at different health cares Data surveying patient preferences from NICE.The preferring Reorting Items for sytemtic and Metananlyses (Prisma) statement guidelines were used followding the manuscript of this review. A protocol was developed and aproved in the PROSPER Database (CRD: 42022327200) Literture search was carried out in September 2022 to idenifiy piblished studies MedMedline, Ambase and Scopus relevant detrurosor acvtivity, ovaractive bladder, urinary incontinence , perspectives, expectationsand preferences. 2 independent authors screened the databses and diesgreementswere resolved upon consensus with a senior author scopeing the review.Studies were included if published in English and available online but excluded if related to children population fecal incontinence postradical post prostatectomy in patients incontinence and in case were not the outcome.The research yielded 1349 papers.Evidence Synthesis asked for evaluation of patients´ prefreneces and ecpetation.Qulittative data evaluated and compared patients preferences and excpeectationsOverall seven studies used a discrete choice system. Patients preferences and expectations in medical treatment 8 studies adressed the the prefrences and expectations of OAB patients, 4 related the use of antimuscarinics, 4 related the treatment unresponsive to antimuscarinics . One study evaluated the importance of economic burden in patinet´s preferences.The retrieved studies highlight the absence of validated and standradized toolsmeasure pts preferences . 1 study evaluated the treatment preferences individuals with symptoms.Treatment-naive pts. Looking at the data currently published shows most of the studies using different non-standardized methods. This may be an issue to change. Patient preferences and excpecations should be a part of the decision making process in benign and malignant disoreders. Malde summarizedpatients preferences and expectations in LUTS management and clearly showed the differenec betwenn doctors and patient wishes and showing that patient were in favour of less effective treatment with minor risk. This study shows a similar confirms this observation. Several explanations hav ebeen shown as a lack of efficacy, rare adeverse events and dpatient awareness of on their condition. A better knowledgeof patinet expectation and preferences can help better help physician to identify the best single patient, to improve patient adherence and compliance.OAB patinet prefere oral medicationreucing urgency, frequency, and incontinence episodeswithout effect of cognitove function and coveres insurance limiting patienta dherence and compliance. Thelack of standardized methods on evaluation reprensent and an important bias in that research area. Sharing desissions on treatment is a corner stone of Treatment sucess, adherence and willingness to treament in OAB patients.
Reply. No reply is due. We thank the reviewer for taking time to review our manuscript.